# Recent Development in Sludge Biochar-Based Catalysts for Advanced Oxidation Processes of Wastewater

**Xingxing Chen** [1,2,3,†], **Liya Fu** [1,2,†], **Yin Yu** [1,2], **Changyong Wu** [1,2,*], **Min Li** [1,2], **Xiaoguang Jin** [1,2,4], **Jin Yang** [1,2,5], **Panxin Wang** [1,2] and **Ying Chen** [3,*]

1   State Key Laboratory of Environmental Criteria and Risk Assessment, Chinese Research Academy of Environmental Sciences, Beijing 100012, China; cxx_scu@163.com (X.C.); liyafu1115@163.com (L.F.); yuyinphoebe@163.com (Y.Y.); lmliminmin@163.com (M.L.); jxg18@mails.tsinghua.edu.cn (X.J.); oyking128@126.com (J.Y.); wangpanxin_hit@126.com (P.W.)
2   Research Center of Environmental Pollution Control Engineering Technology, Chinese Research Academy of Environment Sciences, Beijing 100012, China
3   School of Architecture and Environment, Sichuan University, Chengdu 610065, China
4   School of Environment, Tsinghua University, Beijing 100084, China
5   School of Chemical and Environment Engineering, China University of Mining and Technology, Beijing 100083, China
*   Correspondence: wucy@craes.org.cn (C.W.); chenying@scu.edu.cn (Y.C.)
†   These authors contributed equally to this work.

**Abstract:** Sewage sludge as waste of the wastewater treatment process contains toxic substances, and its conversion into sludge biochar-based catalysts is a promising strategy that merges the merits of waste reutilization and environmental cleanup. This study aims to systematically recapitulate the published articles on the development of sludge biochar-based catalysts in different advanced oxidation processes of wastewater, including sulfate-based system, Fenton-like systems, photo-catalysis, and ozonation systems. Due to abundant functional groups, metal phases and unique structures, sludge biochar-based catalysts exhibit excellent catalytic behavior for decontamination in advanced oxidation systems. In particular, the combination of sludge and pollutant dopants manifests a synergistic effect. The catalytic mechanisms of as-prepared catalysts in these systems are also investigated. Furthermore, initial solution pH, catalyst dosage, reaction temperature, and coexisting anions have a vital role in advanced oxidation processes, and these parameters are systematically summarized. In summary, this study could provide relatively comprehensive and up-to-date messages for the application of sludge biochar-based catalysts in the advanced oxidation processes of wastewater treatment.

**Keywords:** sludge; catalyst; wastewater treatment; advanced oxidation processes; application





## 1. Introduction

With the rapid development of industrialization, urbanization and population, water pollution has become an increasingly concerning issue worldwide [1]. Biological treatments were applied in the wastewater treatments worldwide and a significant amount of waste sludge was produced [2]. The annual output of sludge reached 60–90 million tons in 2020 in China alone [3]. Since sludge contains many toxic substances (such as organic compounds and heavy metals), many conventional sludge disposal methods cause environmental risks; counting landfill, ocean dumping, agriculture usage and incineration are all limited by potential risks [4,5]. In addition, the cost of standardized disposal for the sludge is high, accounting for 30–60% of the total operation costs of wastewater treatment plants [6]. At present, the safe treatment and disposal of sewage sludge pose a great challenge to environmental development.

In recent decades, an ever-increasing variety of organic pollutants in water, such as pharmaceuticals [7,8], pesticides [9], azo dyes [10] and so forth, pose a severe threat to indi-

vidual health and ecosystems. However, conventional wastewater treatment methods such as biological reactors may be inefficacious, arising from low biodegradation efficiency [11]. Up to now, due to the high oxidation potential of the reactive oxygen species (ROS) (i.e., $SO_4\bullet^-$ and $\bullet OH$), the advanced oxidation processes (AOPs) have been widely utilized to remove refractory pollutants from wastewaters [12–14]. Although homogeneous catalytic oxidation technologies are prevalent, transition metals such as $Co^{2+}$, $Fe^{2+}$ and $Mn^{2+}$ are versatile homogeneous catalysts due to their redox potential. However, homogeneous catalytic oxidation technologies generally show the drawback of hardly recycling catalysts. For the sake of more thorough degradation of contaminants and a practicable application, heterogeneous catalysts are supposed to be developed [15].

For achieving "trash-to-treasure", various types of carbon-based catalysts originated from carbon-rich solid waste have been paid great attention recently, such as maize straw, maize cob [16], waste tea leaves [17] and sugarcane residue [18] owing to their rich surface oxygen-containing functional groups (OFGs), high specific area and low cost [1,19]. Meanwhile, sewage sludge is also recognized as a resource of carbonaceous materials to be converted into sludge biochar-based catalysts for heterogeneous advanced oxidation [8,20]. However, few reviews focus on AOPs involving sludge biochar-based catalysts to date.

To better understand the recent development of sludge biochar-based catalysts, this study aims to systematically summarize the application of sludge biochar-based catalysts in different AOPs, including sulfate-based AOPs, Fenton-like AOPs, photocatalysis and ozonation. Additionally, heterogeneous catalytic mechanisms in the reaction systems are taken into account. Furthermore, the effects of various conditions (i.e., sludge components, synthetic approaches, dopants) for as-prepared catalysts on catalytic activity were also introduced. Finally, the prospects and advanced investigations in this field are suggested.

## 2. Preparation of Sludge Biochar-Based Catalysts

### 2.1. Sludge Components

Zaker et al. [5] categorized sludge components into several groups, including non-toxic organic carbon, components containing nitrogen and phosphorus, heavy metals, organic compounds (e.g., PCBs, PAHs) and inorganic compounds (e.g., silicates, aluminates, compounds containing calcium and magnesium), etc. Certainly, sludge components themselves can greatly affect the property of the prepared sludge biochar-based catalysts. For example, lots of hydrophobic phenolic compounds and polycyclic aromatic hydrocarbons (PAHs) exist in coking wastewater treatment sludge, characterized by substituted aromatic structures that are generally the precursors (or intermediate components) for graphitic or polyaromatic carbon generation during sludge pyrolysis [21]. Meanwhile, high specific area and pore volume could be formed due to the volatilization of organic matters. The inorganic components of sludge, mainly formed by elements Si, Al, Mg, Fe, and Zn, all contribute to the catalytic activity of sludge biochar-based catalysts [22]. On one hand, they can be used as inorganic templates. Taking $TiO_2$ as an example, architecture stability and efficiency of light capture of the sludge-supported $TiO_2$ photocatalysts are enhanced compared to $TiO_2$ [23]. On the other hand, inorganic transition metals, such as Fe, Mn and Co, can not only act as catalytic sites [24] to improve catalytic efficiency but also facilitate the formation of persistent free radicals (PFRs) due to their higher redox potential [19] in the preparation process.

### 2.2. Preparation Methods

#### 2.2.1. Pyrolysis

Pyrolysis is one of the most popular methods for the preparation of sludge biochar-based catalysts [24], where sludge biomass can be converted in the absence of oxygen at 300–900 °C [25]. In general, the pyrolysis of sludge can lead to less toxicity due to the reduction of some organic compounds [26] and the immobilization of metals [27]. As far as pyrolysis mechanism was concerned, the online or in situ detection and monitoring techniques such as Pyrolyzer-gas chromatography/mass spectrometry (Py-GC-MS) and

thermogravimetry-Fourier transform infrared spectrometry (TG-FTIR) can provide direct real-time messages including the conversion pathways of functional groups and the formation of pyrolysis products during pyrolysis process [28,29]. In other words, they can further help to understand the sludge pyrolysis mechanism.

The influence factors of pyrolysis process for sludge biochar-based catalysts are mainly involved with pyrolysis temperature, reaction residence time, pyrolysis atmosphere, and so on. Among all parameters, pyrolysis temperature is the most significant factor [1]. Firstly, pyrolysis temperature exerts a significant influence on the physicochemical properties of sludge biochar-based catalysts. Generally, pyrolysis at relatively high temperature tends to cause higher Brunauer–Emmett–Teller (BET) area and total pore volume due to the complete carbonization and the volatilization of organic matter. For instance, the Brunauer–Emmett–Teller (BET) area and total pore volume of magnetic nitrogen-doped sludge-derived biochar (MS-biochar) increased as the pyrolysis temperature rose from 400 °C to 800 °C [30]. However, high-temperature may also incur adverse effects on the BET area and total pore volume, which probably attributed to significant cracking and a decrease in the number of active products [31,32]. Besides, the BET area and pore volume of the sludge biochar-based catalysts could be significantly improved via chemical activations such as KOH, NaOH, and $ZnCl_2$ [24,33,34].

In addition, the organic moieties of sludge biochar-based catalysts are highly pyrolytic temperature dependent. As mentioned above, the conversion of functional groups will take place in the pyrolysis process and the catalytic activity can be impacted by these functional groups. Zhang et al. [21] found that biochars produced from sludge at 700 °C and 900 °C possessed the best catalytic performance, which was ascribed to oxygen-bonded carbon formed at higher pyrolysis temperature. The ratio of organic moieties is also associated with pyrolysis temperature. Wang et al. [31] reported that the content of oxygen-containing functional groups increased when elevating temperature to 600 °C, while it decreased with temperature further increasing. Meanwhile, they pointed out that the resulting conformational change may result from organic moieties-ash interaction, and further manifested that ash, especially the phase transfer of iron oxides, affected the properties of organic moieties during sludge pyrolysis (Equation (1)). It can be believed that the evolution of different metal species relies on pyrolysis temperature. Taking Fe species as a typical example, Zhu et al. [35] showed that the iron species firstly transformed to ferric oxides and then gradually converted to $Fe^0$ along with increasing temperature, and $Fe_3C$ was obtained at 900 °C. Consequently, iron sludge-derived magnetic $Fe^0/Fe_3C$ catalyst exhibited excellent catalytic activity. Yet the iron species can also gradually transform from $Fe_3O_4$ to FeO with increased pyrolysis temperature from 600 °C to 1000 °C [36]. In conclusion, Fe species will undergo a series of phase transformations (high-valence iron to low-valence iron) as anaerobic carbothermal reaction proceeds, and eventually iron compounds with multiple valences may coexist in sludge biochar-based catalysts. Besides, PFRs of sludge biochar-based catalysts are also concerned with pyrolysis temperature. PFRs of sludge biochar-based catalysts can facilitate the formation of ROS and sulfate radicals [1]. Chen et al. [37] found that PFRs concentrations in sludge biochars were remarkably reduced when pyrolysis temperature increased from 400 °C to 1000 °C, but the biochar obtained at 1000 °C still retained excellent catalytic properties.

$$Fe_2O_3 + C_{\text{Reductive form}} \rightarrow Fe_3O_4 + C_{\text{Oxidative form}} \tag{1}$$

When it came to pyrolysis atmosphere, Huang et al. [32] found that sludge biochars formed under the $N_2$, Ar, and $NH_3$ atmospheres all exhibited a high catalytic activity toward peroxymonosulfate (PMS) for rapid removal of bisphenol A (BPA). That is to say, the effect of these different pyrolysis atmospheres was negligible. Moreover, pyrolysis treatment under protective gas may be an effective measure to achieve sludge biochar-based catalysts regeneration.

Until now, microwave pyrolysis technology has drawn growing attention owing to its volumetric, fast, selective and efficient heating [25]. Huang et al. [38] revealed that sewage

sludge biochar under higher microwave power levels can provide better catalytic effect for trichloroethylene in the Fenton oxidation system, probably on account of its higher iron content and specific surface areas. However, microwave pyrolysis technology still faces some practical challenges, such as high energy input, which limits its further application.

### 2.2.2. Hydrothermal Carbonization

Hydrothermal carbonization (HTC) has also emerged as a promising technology to turn sludge into sludge biochar-based catalysts, which usually happens at 180–300 °C and 2–10 MPa for a short duration under an inert atmosphere [39]. The HTC process mechanism of sludge can be obtained via reaction pathways about the char structural transformation of three model compounds (i.e., cellulose and glucose, lignin, and proteins). As Figure 1 presented, more benzene rings, polyaromatic char and N-containing compounds are formed during the HTC process. Compared to pyrolysis, HTC can deal with high moisture sludge without pretreatment and thus save energy, meanwhile, the prepared biochar by HTC has higher carbon content [40]. Certainly, HTC itself has disadvantages as well. For example, there are amounts of soluble organic compounds still left in the aqueous phase after HTC of sludge [41,42].

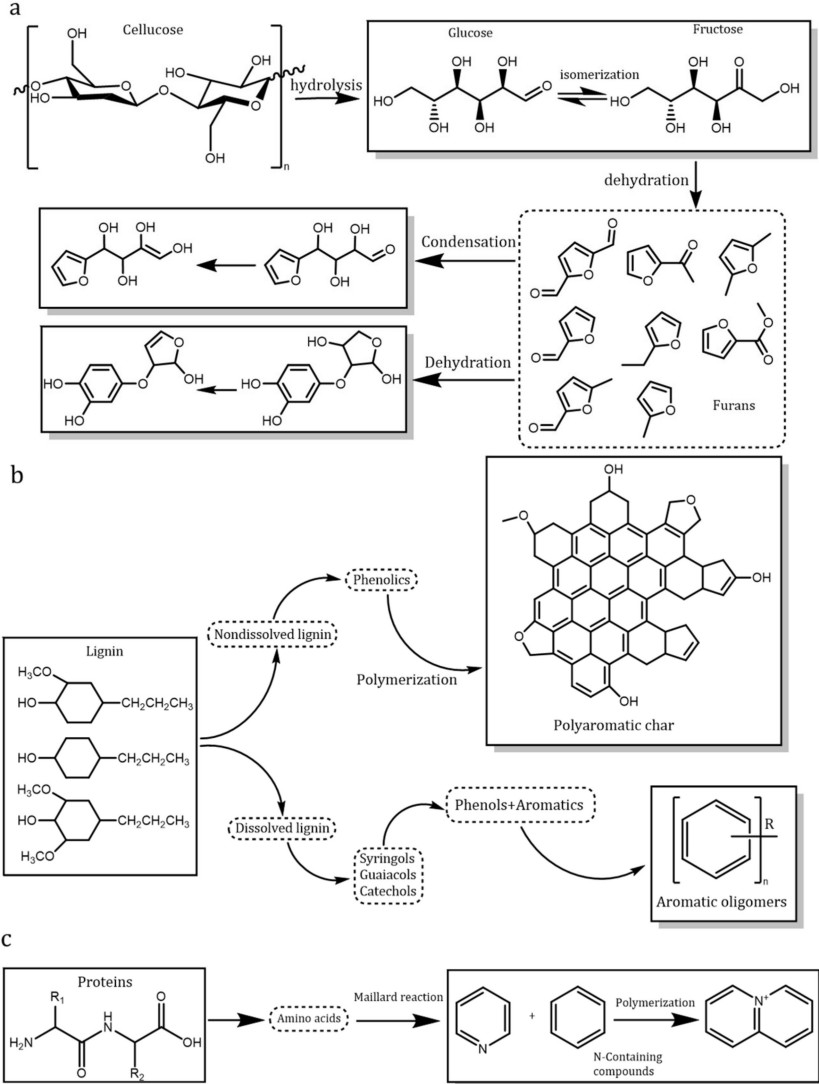

**Figure 1.** (**a**) Reaction pathways of cellulose and glucose decomposition during the HTC process; (**b**) Reaction pathways of lignin during the HTC process; (**c**) Reaction pathways of proteins during the HTC process [39].

Some factors should be taken into consideration when synthesizing sludge biochar-based catalysts using the HTC method, including reaction temperature, pressure, time and so on. Xu et al. [41] revealed that dehydration was the main reaction pathway of sludge during HTC. Correspondingly, dewaterability could be considerably enhanced by elevating reaction temperature in the HTC process [43]. Additionally, reasonable hydrothermal conditions play a crucial role for catalysts in degrading contaminants. Zhang et al. [42] produced magnetic biochar catalysts derived from biological sludge and ferric sludge using hydrothermal carbonization. When applied to treat methylene blue (MB), the MB degradation efficiency reached a plateau along with the temperature rising from 433 K to 473 K or the hydrothermal time increasing from 2 h to 6 h. For MB degradation performance and economic cost, the optimal hydrothermal temperature and time were determined to be 473 K and 6 h. In summary, suitable preparation conditions are of vital importance for sludge biochar-based catalysts during HTC.

### 2.3. Dopants

When sewage sludge components are insufficient for the property of pure sludge biochar catalysts, it is essential to integrate other elements [24]. Generally, the doping of metals and heteroatoms is expected to act as a catalytic site and subsequently enhance catalytic activity of sludge biochar-based catalysts.

The significant amount of transitional metals such as Fe, Mn and Co produce various metal phase structures in sludge biochar-based catalysts. In particular Fe metal phases (such as $\alpha$-$Fe_2O_3$, $Fe_3O_4$, $Fe_3C$ and $Fe^0$, etc.) could show magnetic behavior to easily recycle. Furthermore, the HTC process is prone to $Fe_3O_4$ crystal growth in the presence of reductants, such as biosolids [44]. Therefore, Fe doping sludge biochar-based catalysts are commonly used in Fenton-like or sulfate-based AOPs. In addition, incorporating transitional metals, mainly Ti, in sludge biochar-based catalysts for photocatalysis is conducive to activating the photosensitivity of composites. On the whole, the addition of metals will serve as active sites in AOPs. However, excessive metal loading would reduce the density of active sites on the surface of catalysts and destroy the regular structures [34]. Thus, the catalytic capacity of sludge biochar-based catalysts greatly weakened. Moreover, the presence of transitional metals is an important factor affecting the generation of ROS and PFRs in sludge biochar-based catalysts.

Among heteroatoms doping in sludge biochar-based catalysts, N doping is preferential due to its comparable atom size and strong bond with carbon atoms [45]. N species such as pyridinic N and graphitic N are beneficial to enhance catalytic activity [45,46]. For sulfate-based AOPs, the doping of nitrogen can substantially alter the catalytic oxidation pathway from a radical process to a nonradical process [47]. The doping of nitrogen can also improve the persulfate activation capacity of sludge biochar-based catalysts via enhancing the electron shuttling capacity in nonradical pathways [48]. Mian and Liu [49] found that a non-radical process performed by pyridinic N was the dominant catalytic mechanism in the sludge biochar/PMS system. In addition, pyrrolic N was also found to favor the adsorption of pollutants. In a photocatalysis system, N doping can form graphitic carbon nitride (g-$C_3N_4$) so that photocatalytic capacity of sludge biochar-based catalysts promotes dramatically. Besides, quaternary-N can act as active sites for activation of $H_2O_2$ [50]. Nevertheless, too much doping of N may change the nature of catalysts and introduce new pollutants [51].

### 3. Influence Parameters in AOPs

Influence parameters in AOPs contain initial solution pH, catalyst dosage, reaction temperature, and coexisting anions, to summarize the data in Table 1.

**Table 1.** Influence parameters in AOPs.

| Parameters | Catalysts | Removal Capacity (RC) | AOPs | References |
|---|---|---|---|---|
| pH = 3.2<br>pH = 7.2<br>pH = 10.2 | SBC | RC(triclosan)~70%<br>RC(triclosan) = 99.2%<br>RC(triclosan) = 46.7% | sulfate-based AOPs | [52] |
| pH = 4 | Fe-SC | RC(AOII) = 99% | Fenton-like | [53] |
| pH = 3.1 | BC | RC(trichloroethylene) = 83% | Fenton-like | [38] |
| pH = 7<br>pH = 11 | $MnO_x$/SBAC | RC(COD)~65%<br>RC(COD)~75% | ozonation | [54] |
| Catalyst dosage:<br>0.1 g/L<br>0.3 g/L<br>1 g/L | SBC | RC(triclosan) = 16%<br>RC(triclosan) = 67%<br>RC(triclosan)~100% | sulfate-based AOPs | [52] |
| Catalyst dosage:<br>0.5 g/L<br>0.9 g/L<br>1.1 g/L | ZnO@RSDBC | RC(AO7) = 64,7%;<br>RC(AO7) = 93.7%;<br>RC(AO7) = 95.9% | photocatalysis | [10] |
| T = 15–35 °C | SBC | The degradation rate of triclosan was increased | sulfate-based AOPs | [52] |
| T = 35 °C, 25 °C, 15 °C<br>and 5 °C | Co-Fe/$SiO_2$ LC | The degradation rate constants of ciprofloxacin for 35 °C, 25 °C, 15 °C and 5 °C were 1.615, 0.686, 0.398 and 0.173 $min^{-1}$ | sulfate-based AOPs | [55] |
| T = 15 °C<br>T = 55 °C | $MnFe_2O_4$-SAC | The degradation rate constants of Orange G for 15 °C and 15 °C were 0.07 and 0.247 $min^{-1}$ | sulfate-based AOPs | [56] |
| T = 20 °C<br>T = 40 °C<br>T = 60 °C | SC-$H_2SO_4$ | RC(ofloxacin) = 23.3%<br>RC(ofloxacin) = 80.4%<br>RC(ofloxacin) = 91.5% | Fenton-like | [57] |
| $HCO_3^-$ | Co-Fe/$SiO_2$ LC | 0.5 mM, there was a slight decrease<br>1–10 mM, the removal of ciprofloxacin was accelerated | sulfate-based AOPs | [55] |
| $HCO_3^-$ | $CoFe_2O_4$-SAC | 0–0.1 mg/L, the removal of norfloxacin was accelerated<br>0.1–0.4mg/L, it inhibited the performance of norfloxacin degradation | sulfate-based AOPs | [8] |
| $CO_3^{2-}$ | SC | 0 mM, RC(norfloxacin) = 99.0%<br>10 mM, RC(norfloxacin) = 65.8% | Fenton-like | [58] |
| $HCO_3^-$ | SBC | 0 g/L, RC(TOC)~85%<br>0.6 g/L, RC(TOC)~70%<br>1 g/L, RC(TOC)~70% | ozonation | [22] |
| $HCO_3^-$ | BC | 0 mM, RC(phenol)~89%<br>0.5 mM, RC(phenol)~91.5%<br>50 mM, RC(phenol)~99.9% | ozonation | [21] |
| $HCO_3^-$ | $MnO_x$/SBAC | 0 g/L, RC(TOC) = 64.4%<br>50 mg/L, RC(TOC) = 50.9% | ozonation | [59] |
| $Cl^-$ | Co-Fe/$SiO_2$ LC | 2–50 mM, the degradation rate was decreased | sulfate-based AOPs | [55] |
| $Cl^-$ | $CoFe_2O_4$-SAC | 0–0.1 g/L, the degradation rate of norfloxacin was reduced<br>0.1–0.4 g/L, the degradation rate was increased | sulfate-based AOPs | [8] |
| $Cl^-$ | ADSBC 1000 | 0–20 mM, there was little effect on sulfathiazole degradation | sulfate-based AOPs | [37] |
| $Cl^-$ | SC | 10 mM, there was no significantly negative effect on the norfloxacin removal | Fenton-like | [58] |
| $Cl^-$ | BC | 50 mM, there was no effect on phenol removal | ozonation | [21] |
| $PO_4^{3-}$ | ADSBC 1000 | 0–20 mM, there was little effect on sulfathiazole degradation | sulfate-based AOPs | [37] |
| $PO_4^{3-}$ | SC | 0 mM, RC(norfloxacin) = 99.0%<br>10 mM, RC(norfloxacin) = 83.1% | Fenton-like | [58] |
| $PO_4^{3-}$ | BC | 0.5 mM, there was no effect on phenol removal | ozonation | [21] |
| $NO_3^-$ | $CoFe_2O_4$-SAC | 0–0.4 g/L, there was weak influence on norfloxacin degradation | sulfate-based AOPs | [8] |
| $NO_3^-$ | ADSBC 1000 | 0–20 mM, there was little effect on sulfathiazole degradation | sulfate-based AOPs | [37] |
| $NO_3^-$ | SC | 10 mM, there was nearly no negative effect on norfloxacin removal | Fenton-like | [58] |
| $NO_3^-$ | BC | 50 mM, there was little effect on phenol removal | ozonation | [21] |

### 3.1. Initial Solution pH

Initial solution pH as an important parameter determines the activity of heterogeneous advanced oxidation processes for pollutants elimination in some respects. In general, the pH of solution affects the surface charge of sludge biochar-based catalysts as well as the speciation of contaminants [35]. Taking triclosan degradation in heterogeneous sulfate-based AOPs as an example, triclosan decay reached 99.2% at pH 7.2 and decreased at alkaline pH 10.2 [52]. Considering dissociation constants of triclosan (pKa = 7.8) and the point of zero charge of sludge-derived biochar catalyst ($pH_{pzc}$ = 7.5), electrostatic repulsion caused the depression of catalytic efficiency in that catalyst surface was negatively charged and triclosan was anion state at pH 10.2. Regardless of extreme alkali pH or extreme acid pH, sludge biochar-based catalysts activating persulfate can work in a wider pH range

with a relatively high decontamination efficiency. Since the generation of reactive oxidizing species is related to solution pH, heterogeneous Fenton-like and catalytic ozonation are more sensitive to pH variation in most cases in contrast.

Generally, acidic solution is beneficial for heterogeneous Fenton-like processes [53], and neutral and alkaline conditions are in favor of heterogeneous catalytic ozonation [54]. This phenomenon also occurs in photocatalysis. The formation of positive holes is in preference to hydroxyl radicals at low pH conditions, while hydroxyl radicals predominate under neutral and alkaline conditions [60]. Besides, solution pH also affects the stability of sludge biochar-based catalysts via the metals leaching. Huang et al. [38] carried out a heterogeneous Fenton-like study, in which copper, iron, and zinc leachability from sewage sludge biochar in various pH values were investigated. The results indicated that the highest copper, iron, and zinc concentrations were found at pH 3, and that metals leaching concentration decreased with increasing pH. We can conclude that more metals leaching is likely to happen at low pH, and this in turn leads to secondary pollution and simultaneously destroys the stability of catalysts. Thus, knowing the optimum pH range can enable us to enhance recalcitrant organics removal rate and promote the reuse of sludge biochar-based catalysts.

### 3.2. Catalyst Dosage

Catalyst dosage is one of the important governing factors with respect to the catalytic efficiency in heterogeneous advanced oxidation processes. It is known that the higher catalyst dosage results in more active sites that are available for the generation of free radicals. Nevertheless, the degradation efficiency may become no significant difference with excessive catalyst dosage. As a matter of fact, high catalyst concentration enhances the scavenging effect of free radicals, especially in the photocatalysis system, and it may lead to aggregation of catalysts and thereby allow for less light transmission [10]. Thus, optimal catalyst dosage should be determined based on both economic assessment and removal efficiency.

### 3.3. Reaction Temperature

Several researchers have investigated the critical role of reaction temperature in heterogeneous sulfate-based AOPs utilizing sludge biochar-based catalysts. For example, Zhu et al. [55] analyzed the effect of temperature (in the range of 5–35 °C) on the degradation of ciprofloxacin. In this study, the degradation rate constants increased with reaction temperature. A positive correlation established between reaction temperature and pollutant removal was also reflected in the decay of triclosan [52] and Orange G [56] in the same system. This was attributed to faster activation of persulfate and more formation of reactive radicals at higher temperature. A similar trend was observed in heterogeneous Fenton oxidation. Yu et al. [57], in an ofloxacin degradation study, changed reaction temperature from 20 °C to 60 °C to establish the optimum value of reaction temperature. They reported that the highest ofloxacin decay and $H_2O_2$ decomposition occurred at 60 °C, although reaction temperature of Fenton-like processes was typically located around 25–30 °C [61]. The above results could be perfectly explained with the increase of kinetic constants within a certain temperature range, according to the Arrhenius law [62]. Moreover, reaction temperature also has an optimum point in the heterogenous photocatalytic process. At higher temperature, faster reaction rates improved the degradation efficiency, while less adsorption of contaminants on the carriers made the degradation efficiency unfavorable [60]. It was noteworthy that the optimal temperature of heterogeneous catalytic ozonation was generally found at 25 °C in most cases [63].

### 3.4. Coexisting Anions

Numerous anions in the real wastewater matrix, such as $CO_3^{2-}$, $HCO_3^-$, $Cl^-$, $PO_4^{3-}$ and $NO_3^-$, are vital elements for heterogeneous advanced oxidation processes as they may react with reactive radicals and inevitably affect the catalytic efficiency [14,64].

Chen et al. [22] found that $HCO_3^-$ can function as a strong •OH scavenger in heterogeneous catalytic ozonation system. They affirmed that •OH-mediated oxidation was a main mechanism of waste sludge biochar catalytic ozonation in light of the bicarbonate quenching experiment. On the contrary, $HCO_3^-$ could exert a positive effect on catalytic ozonation. As Zhang et al. [21] observed, the presence of $HCO_3^-$ enhanced the phenol removal. The reason for this result may be that $HCO_3^-$ improved the formation of •$O_2^-$ radical species that were deemed as the main promoter for phenol degradation in the sludge biochar/$O_3$ system. On the other hand, Zhuang et al. [59] found that 50 mg/L $HCO_3^-$ had little effect on the TOC removal of Lurgi coal gasification wastewater, which was mainly due to the production of $H_2O_2$. Similarly, $CO_3^{2-}$/$HCO_3^-$ would bring about inhibition, enhancement or dual influence on other heterogeneous catalytic processes as well. A study was conducted to examine the impact of $CO_3^{2-}$ on Fenton-like reactions [58]. The alkaline solution pH caused by 10 mol/mL $CO_3^{2-}$ addition, as mentioned earlier, probably impeded the heterogeneous Fenton-like reactivity. $CO_3^{2-}$ and $HCO_3^-$ are able to activate persulfate for pollutant deterioration [14] and, in the meantime, they are capable of clearing away $SO_4$•$^-$ and •OH radicals. As a consequence, it was observed that the effect of $CO_3^{2-}$/$HCO_3^-$ on sulfate-based AOPs to some extent relied on their concentration in aqueous solution. As Zhu et al. [55] showed, the ciprofloxacin removal was slightly diminished when 0.5 mmol/L $HCO_3^-$ was added, but nevertheless, the ciprofloxacin decay was significantly elevated by a rise of $HCO_3^-$ concentration (1–5 mmol/L). Surprisingly, the degradation rate of ciprofloxacin was highly boosted as the amount of ciprofloxacin increasing to 10 mmol/L. Apart from the above reasons, the positive result may be also owing to the forming of reactive metal carbonate complexes at higher $HCO_3^-$ contents. A different phenomenon was found regarding the norfloxacin (NOR) degradation, where a lower $HCO_3^-$ concentration resulted in a positive effect and a relatively higher $HCO_3^-$ concentration, reversed [8].

Several investigations have indicated that $Cl^-$ ion makes a difference of more or less to HO•$^-$ and $SO_4$•$^-$-based AOPs [65]. $Cl^-$ is able to consume HO• and $SO_4$•$^-$ and then produce Cl•$^-$ and $Cl_2$•$^-$ radicals with lower redox potential. Thus, $Cl^-$ may incur an inhibitory effect. For example, the ciprofloxacin removal rate exhibited a slight drop under the studied $Cl^-$ concentration (2–50 mmol/L) [55]. However, it has not been always observed and sometimes the presence of $Cl^-$ will accelerate the abatement of contaminants. Yang et al. [8] reported that detrimental effects appeared at low concentrations of $Cl^-$ (0.05–0.1 g/L), but further addition of $Cl^-$ (0.1–0.4 g/L) comparably enhanced the degradation of NOR. Improvement at a relatively high concentration of $Cl^-$ concentration can be understood in some respects. The formation of reactive halogen under excessive $Cl^-$ conditions, such as HOCl and $Cl_2$, partly accounts for this outcome [8,51]. Another explanation for this behavior is perhaps that $Cl^-$ is capable of activating PMS [52]. Besides, as has been already reported, $Cl^-$ plays a crucial part in the Fenton oxidation [66,67]. That notwithstanding, $Cl^-$ may also induce no evident alteration of degradation efficiency in the heterogeneous AOPs [21,37,58].

Similarly, $PO_4^{3-}$ is pervasive in wastewater and holds a debatable role in the heterogeneous AOPs. As far as $NO_3^-$ ion is concerned, $NO_3^-$ is generally reported to have a mild effect on catalytic oxidation [68].

## 4. Application in AOPs

### 4.1. Application in Sulfate-Based AOPs

In recent years, sulfate-based AOPs have received a great deal of attention in water treatment because of $SO_4$•$^-$ [7]. Compared with its counterpart •OH, $SO_4$•$^-$ has a higher oxidation potential (2.5–3.1 V), a wider pH range, a longer half-life (30–40 μs) and a higher selectivity for target contaminants [7,55]. It is well known that $SO_4$•$^-$ can be generated by activating PMS or peroxodisulfate (PDS) with transition metals and catalysts and so on. Sludge biochar-based catalysts as the activator of PMS/PDS for wastewater treatment have also been widely studied (Table 2).

**Table 2.** Application of sludge-based catalysts in sulfate-based AOPs.

| Synthesis Process | Product | Solution pH | Removal Capacity (RC) | Reusability and Chemical Stability | Mechanism | References |
|---|---|---|---|---|---|---|
| Iron sludge + ethylene glycol + $CoCl_2$ + NaAc were vigorously stirred, then the suspension was solvothermal treated at 200 °C for 10 h. | Co-Fe/$SiO_2$ LC | 7.0 | 0.2 g/L product; 10 mg/L ciprofloxacin; 0.5 g/L PMS. RC(ciprofloxacin) > 99.6% | RC(ciprofloxacin) decreased to 72.0% (4th run). | $SO_4\bullet^-$, $\bullet OH$ | [55] |
| The dried sludge was calcined at 450 °C for 0.5 h under $N_2$ and the obtained biochar was further activated by NaOH. | SBC | 7.2 | 0.5 g/L product; 0.034 mmol/L TCS; 0.8 mmol/L PMS. RC(TCS) = 99.2% | After 5 rounds, RC(TCS) decreased to 53.5%. | $SO_4\bullet^-$, $\bullet OH$, $^1O_2$ | [52] |
| The dried sludge was carbonated at 600 °C for 6 h under $NH_3$/Ar. | Biochar | 6 (phosphate buffer) | 0.2 g/L product; 10 ppm BPA; 0.1 g/L PMS. RC(TOC)~80% | - | $^1O_2$ | [32] |
| ADS was annealed at 1000 °C for 90 min under $N_2$. | ADSBC 1000 | 6 | 0.5 g/L product; 20 mg/L STZ; 10 mmol/L PDS. RC(STZ) = 90.31% | - | Nonradical process | [37] |
| The centrifuged sewage sludge + iron salt mixture was treated at 180 °C for 3 h in $N_2$. | IBHC | 4 | 0.2 g/L product; 60 mg/L tetracycline; 5 mmol/L PDS. RC(tetracycline) = 99.72% | RC(tetracycline) was 94.7% in the fifth round reuse. | $SO_4\bullet^-$, $\bullet OH$ | [69] |
| The preprocessed iron sludge was annealed at 900 °C for 2 h under Ar. | $Fe^0$/$Fe_3C$@C900 | 7.0 | 0.2 g/L product; 10 mg/L ciprofloxacin; 0.50 g/L PMS. RC(ciprofloxacin) = 98.2% | For the third run, 99% of ciprofloxacin removal was achieved and the concentration of leached Fe decreased from 0.684 to 0.227 mg/L. | $SO_4\bullet^-$, $\bullet OH$, $O_2\bullet^-$, $^1O_2$ | [35] |
| The dried sludge was treated by $NaBH_4$ and then pyrolyzed at 400 °C for 2 h under $N_2$. | ZVI-SDBC | 5.22 | 0.5 g/L product; 0.06 mol/L AO7; 0.925 mmol/L PDS. RC(AO7) = 99.0% | The rate constants were 0.0718, 0.0655 and 0.0502 $min^{-1}$ in the first-cycle, second-cycle and third-cycle reuse. | $SO_4\bullet^-$, $\bullet OH$, $^1O_2$ | [70] |
| The sludge granule was pyrolyzed at 600 °C for 2 h (SDBC). Then, the SDBC + $MnCl_2$ mixture was pyrolyzed at 600 °C for 30 min. | Mn-SDBC | 6 | 2 g/L product; 1500 mg/L OG; 3 mmol/L PDS. RC(OG) = 95.94% | | $SO_4\bullet^-$, $\bullet OH$ | [71] |
| The dried sludge + urea mixture was calcined at 700 °C under $N_2$ for 2 h. | NC-700 | - | 0.3 g/L product; 50 mg/L MB; 0.4 g/L PMS. RC(MB) = 98.7% | >95% of MB could be removed over five cycles. | $^1O_2$ | [68] |
| The urea + sludge mixture was calcined at 550 °C for 2 h. | UBC-0.5 | 6.84 | 0.5 g/L product; 0.1 mmol/L BPA; 1 mmol/L PMS. RC(BPA)~100% | A slight decrease was observed. | $^1O_2$ | [51] |
| The dried sludge powders were pyrolyzed at 800 °C for 2 h under $N_2$. | MS-800 | 2.17 | 0.2 g/L product; 10 mg/L tetracycline; 4.2 mmol/L PDS. RC(tetracycline) = 82.24% | The Fe highest leaching concentration was 0.8 mg/L. MS-800 exhibited a better reusability after four times application. | $SO_4\bullet^-$, $\bullet OH$ | [30] |
| Sludge + agar powder + $MnCl_2$ + $NH_4OH$. Dried solid was thermal treated at 800 °C for 1 h under Ar. | ASMn-Nb | 6 (phosphate buffer) | 0.2 g/L product; 20 mg/L AO7; 1.6 mmol/L PMS. RC(AO7) = 100% | After 5 times recycling, the complete degradation of AO7 was achieved and there was negligible metal leaching. | Radical and Nonradical process | [46] |
| SAC + Co $(NO_3)_3 \cdot 9H_2O$ + Fe $(NO_3)_3 \cdot 9H_2O$. Then NaOH solution was added until pH = 12. The suspension was treated at 180 °C for 12h. | $CoFe_2O_4$-SAC | - | 0.1 g/L product; 10 mg/L NOR; 0.15 g/L PMS. RC(TOC) = 81.0% | RC(NOR) maintained at 90% after five cycles. The leaching concentration of cobalt and iron was 0.57 and 0.25 mg/L, respectively. | $SO_4\bullet^-$, $\bullet OH$ | [8] |
| SAC + $FeCl_3 \cdot 6H_2O$ + $MnCl_2 \cdot 4H_2O$ were dissolved in ethylene glycol under ultrasonication. Later, NaAc was added and stirred. Finally, the mixture was treated at 200 °C for 10 h. | $MnFe_2O_4$-SAC | - | 0.2 g/L product; 20 mg/L OG; 0.5 g/L PDS. RC(OG)>95% | RC(OG) was more than 94%, even after five cycles | $SO_4\bullet^-$, $\bullet OH$ | [56] |
| Magnetic porous carbon was microwave digested and carbonized at 600 °C for 2 h under $N_2$. | MS600 | 7.0 | 1 g/L product; 1 mmol/L 2-Napthol; 20 mmol/L PDS. RC(2-Napthol) = 88.7% | After the third time, RC(2-Napthol) was still above 80%. | $SO_4\bullet^-$, $\bullet OH$ | [31] |

### 4.1.1. Pure Sludge-Derived Biochar Catalysts

Pure sludge-derived biochar catalysts can be used as activators towards persulfate due to their oxygen-containing functional groups such as semiquinone and specific sludge components [52]. For example, sludge biochar with excellent catalytic performance to-

wards PMS was synthesized by Huang et al. [32], wherein metals in the sludge promoted catalytic activity of biochar in pyrolysis and the ketone structure inside the biochar also contributed to producing $^1O_2$. Apart from this radical process, the direct electron transfer without formation of radicals (i.e., nonradical pathway) also plays a role in persulfate activation processes. The mediated electron transfer could act as an activator of PMS when utilizing carbonaceous materials, such as single-walled carbon nanotubes (electron transfer mediator) [72]. Comparatively, sludge-derived biochars can also act as a mediation for electron transfer from pollutants (electron donor) to persulfate (electron acceptor) in a nonradical pathway as a result of their high graphitization degree and conductivity [37], which is surely responsible for persulfate activation.

The catalytic ability of pure sludge-derived biochar in sulfate-based AOPs was investigated. Wang and Wang [52] applied pure sludge-derived biochar (SBC) compared with the Fe(II)/PMS system to activate PMS for the degradation of triclosan. Based on experimental results, the removal efficiency of triclosan in the SBC/PMS system (99.2%) was higher than that of Fe(II)/PMS system (98.9%) under each optimum condition. Compared to simulated wastewater, triclosan removal efficiency decreased in actual wastewater, indicating that wastewater components had a negative effect on triclosan degradation. In addition, the kinetics of PMS activation by sludge-derived biochar followed the two-phase kinetic model, i.e., fast phase and slow phase [73]. The fast phase is mainly attributed to the formation of surface-bound reactive species. Moreover, organic pollutants can affect the PMS activation kinetics by competing for the adsorption sites with PMS, depending on the physicochemical properties of organic pollutants.

### 4.1.2. Fe-Based Catalysts

Among transition metals, iron salt has widely been used for persulfate activation based on its low-cost, nontoxicity, convenient operation and very fast reaction rate [74]. Wei et al. [69] produced modified sludge bio-hydrochar (IBHC) by one-pot hydro-carbonization of the mixture of sludge and iron salt $((NH_4)_2Fe(SO_4)_2 \cdot 6H_2O)$. Iron species presented as FeO and FeOOH in light of XPS spectra, which significantly enhanced PDS decay through a series of reactions (Equations (2)–(3)) and thereby efficiently degraded pollutants. Additionally, the catalyst containing oxygen functional groups may act as the electron transfer mediator to generate $SO_4 \bullet^-$, and $SO_4 \bullet^-$ in aqueous can also produce $\bullet OH$ (Equation (4)).

As an alternative activator of Fe(II), zero-valent iron ($Fe^0$, ZVI) can reduce sulfate radical consumption due to excess $Fe^{2+}$ and avoid the introduction of other anions [74]. In the system of $Fe^0$/PMS, $Fe^0$ can activate PMS directly (Equations (5)–(6)) and accelerate the transformation from $Fe^{3+}$ to $Fe^{2+}$ (Equation (7)) [35]. However, pure $Fe^0$ is easy to agglomerate, and previous studies show that carbon materials as a supporting matrix of $Fe^0$ can greatly overcome this drawback [71]. Iron sludge-derived magnetic $Fe^0$/$Fe_3C$ catalyst carbonized at 900 °C under Ar atmosphere presented excellent catalytic activity for ciprofloxacin degradation by PMS activation, where $SO_4 \bullet^-$, $\bullet OH$, $O_2 \bullet^-$ and $^1O_2$ all contributed to the decomposition of ciprofloxacin [35]. According to characterization results, the cladding of carbon strengthened its reusability and stability due to the porous carbon spheres and the iron-carbon alloy structure. In view of the aforementioned advantages of ZVI-based carbon catalyst, it may be suitable for environmental remediation. Typically, ZVI-sludge-derived biochar (SDBC) could be obtained through one-step pyrolysis; an SDBC/PDS system was applied to repair landfill leachate [70]. Satisfactorily, the TOC and ammonia removal of leachate reached the high levels of 62.8% and 99.8%, respectively, when experiment conditions were 30.0 mmol/L PDS and 5.0 g/L SDBC.

$$2FeOOH + 2SO_4^{2-} + 2H_2O \rightarrow 2FeSO_4 + 4OH^- + H_2 + O_2 \tag{2}$$

$$Fe^{2+} + S_2O_8^{2-} \rightarrow Fe^{3+} + SO_4 \bullet^- + SO_4^{2-} \tag{3}$$

$$SO_4 \bullet^- + H_2O \rightarrow \bullet OH + HSO_4^- \tag{4}$$

$$Fe^0 + 3HSO_5^- \rightarrow 3 \bullet OH + Fe^{3+} + 3SO_4^{2-} \tag{5}$$

$$Fe^0 + 2HSO_5^- \rightarrow 2SO_4\bullet^- + Fe^{2+} + 2OH^- \tag{6}$$

$$Fe^0 + 2Fe^{3+} \rightarrow 3Fe^{2+} \tag{7}$$

### 4.1.3. Mn-Based Catalysts

With different valences, manganese oxides-loaded sludge biochar-based catalysts have been applied in AOPs, such as heterogeneous catalytic ozonation. Similar to Fe, Mn can also activate PDS to produce $SO_4\bullet^-$ due to redox pairs (Equation (8)). The Orange G degradation by manganese oxide-loaded sludge-derived biochar (Mn-SDBC) for PDS activation reached 34.15% to 90.28% when Mn oxide loading rate increased from 0 to 40 mmol/L [71]. Furthermore, it was also efficient for the Mn-SDBC/PDS system to degrade Orange G under a continuous flow condition in the fixed-bed column.

$$Mn^{2+} + S_2O_8^{2-} \rightarrow Mn^{3+} + SO_4\bullet^- + SO_4^{2-} \tag{8}$$

### 4.1.4. Heteroatom-Doped Hybrid Catalysts

The doping of heteroatoms, especially for nitrogen doping in sludge biochar-based catalysts, will bring about a positive effect. Hu et al. [68] prepared nitrogen-doped sludge carbon catalyst through one-pot pyrolysis. Upon the addition of nitrogen, the catalyst took on a strong adsorption ability and high catalytic activity for PMS activation. In this study, radical pathways ($^1O_2$ as main reactive species) dominated organic pollutants degradation in which radical reactions were proposed as shown in Equations (9)–(13). By contrast, non-radical pathways made little contribution. Accordingly, their activation pathways were elucidated in Figure 2.

$$HSO_5^- + H_2O \rightarrow HSO_4^- + H_2O_2 \tag{9}$$

$$H_2O_2 + \bullet OH \rightarrow HO_2\bullet + H_2O \tag{10}$$

$$HO_2\bullet \rightarrow H^+ + O_2\bullet^- \tag{11}$$

$$O_2\bullet^- + \bullet OH \rightarrow {}^1O_2 + OH^- \tag{12}$$

$$2O_2\bullet^- + 2H^+ \rightarrow H_2O_2 + {}^1O_2 \tag{13}$$

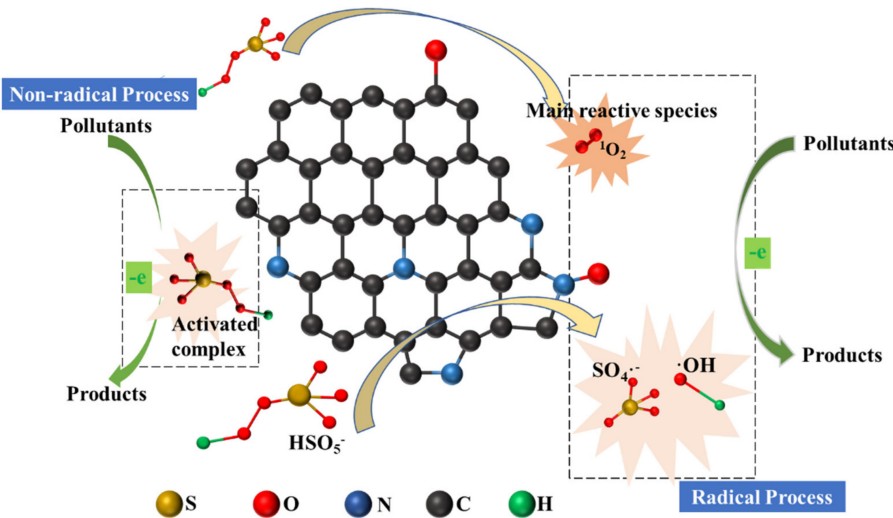

**Figure 2.** Mechanism of PMS activation and pollutants degradation over nitrogen-doped sludge carbon catalyst [68].

Sludge biochar-based catalysts prepared by a combination of both metal and nitrogen can further increase PMS/PDS activation activity and the stability of catalysts. For example, magnetic Fe,N-codoped carbon catalysts (UBC-x, x denotes the mass ratio of urea to dry

sludge) were prepared at 550 °C for 2 h using Fe-rich sludge and N-rich urea as the raw materials [51]. There 75% of BPA could be removed by the Fe-rich sludge-based biochar/PMS system, whereas the removal of BPA reached nearly 100% in UBC-0.5/PMS system at the same conditions. Concerning its superior performance, Fe and N served as active sites and there was no Fe and N leaching during the degradation process. In the meantime, $^{1}O_2$ was the primary reactive species for BPA degradation. Magnetic nitrogen-doped sludge-derived catalysts (MS-biochar) from polyacrylamide-polyferric sulfate-flocculated municipal sewage sludge also reflected excellent PDS activation efficiency [30]. Unlike other studies, MS-biochar was divided into four parts in this study, including dissolved organic matter (DOM), acid-soluble substance (ASS), carbon matrix (CM), and basal part (BSP). The contribution of different parts of MS-biochar for PDS activation was discussed, and it relied on the preparation temperature of MS-biochar. When further elaborating activation mechanisms of MS obtained at 800 °C, the results proved iron-compounds in ASS as catalytic sites devoted to $SO_4\bullet^-$ generation. Meanwhile, both graphitic carbon and doped nitrogen species in CM primarily committed to $\bullet OH$ generation. In particular, the internal electron migration path (from $sp^3$ to nanocrystalline $sp^2$ carbon) was firstly proposed to explain the activation mechanism, as shown in Figure 3.

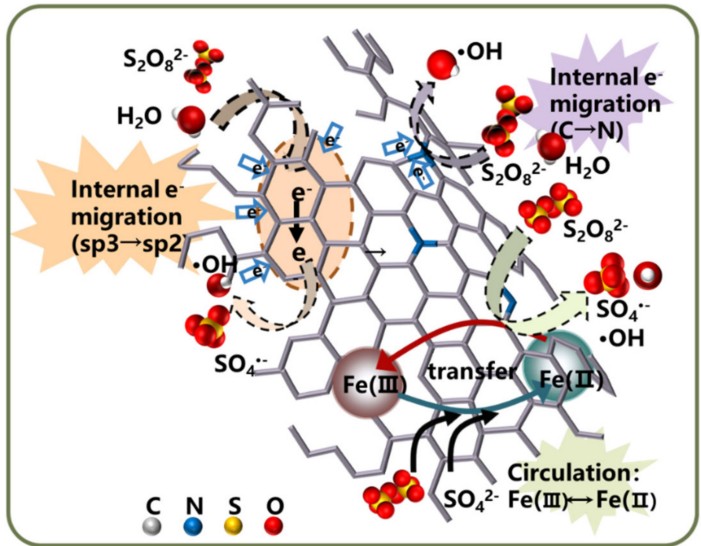

**Figure 3.** Activating mechanism over MS-biochar [30].

In addition to Fe,N-doped composites, sludge-derived MnOx-N-biochar (ASMn-Nb) was also prepared as a catalyst for PMS activation [46]. When catalytic performance of ASMn-Nb was evaluated via acid orange 7 (AO7) and rhodamine B degradation, the kinetics results revealed that most rapid degradation occurred in ASMn-Nb/PMS system, which was consistent with sludge-derived Fe,N-doped composites. Based on N-carbon as the main reactive site, non-radical process, i.e. mediated electron transfer, was the primary mechanism of PMS activation, which can be also verified in another study [49].

### 4.1.5. Multimetallic Catalysts

It has been found that $Co^{2+}$ is the best PMS activator [75–77]. However, the application of $Co^{2+}$ is constrained due to its toxicity. Luckily, magnetic Co-based multimetallic catalysts have also been used in sulfate-based AOPs. For instance, Zhu et al. [55] fabricated magnetic Co-Fe/SiO$_2$ layered catalyst (LC) by reusing iron sludge as the iron source and substrate via a facile hydrothermal method. Coupling Co-Fe/SiO$_2$ LC with PMS was effective for degradation of ciprofloxacin. Within 1 h, the degradation and mineralization efficiency of ciprofloxacin reached over 99.6% and 53.8%, respectively, which was much higher than the commercial Fe$_2$O$_3$-PMS, Fe$_3$O$_4$-PMS, Co$_3$O$_4$-PMS system. Interestingly, the removal efficiency of ciprofloxacin was less than 20% in the Co-Fe/SiO$_2$ LC-PDS system when pH

was 7. This indicated that the asymmetric PMS was easier to activate in sulfate-based AOPs, which may be associated with the lower O-O bond energy of PMS [52]. Meanwhile, the results of the quenching experiments and electron paramagnetic resonance (EPR) tests all confirmed the participation of both $SO_4\bullet^-$ and $\bullet OH$. The Co-Fe bimetallic catalyst enhanced electron transfer between $Co^{2+}$ and $Fe^{2+}$, followed by the decomposition of PMS to produce $SO_4\bullet^-$ and $\bullet OH$ (Equations (14) and (15)). This synergistic effect catalyst may result from the reduction of $Co^{3+}$ by $Fe^{2+}$ (Equation (16)).

The spinel cobalt ferrite ($CoFe_2O_4$) nanoparticles (NPs) are also an example. A novel magnetic $CoFe_2O_4$-SAC composite was obtained by decorating $CoFe_2O_4$ on sewage sludge-derived activated carbon (SAC), and its PMS activation performance was evaluated by the degradation of NOR [8]. Unsurprisingly, the degradation and mineralization efficiencies of the $CoFe_2O_4$-SAC/PMS system were better than those of the $CoFe_2O_4$/PMS system. The introduction of SAC partially prevented the aggregation of magnetic $CoFe_2O_4$ NPs and, as mentioned above, acted as a catalyst via radical/non-radical pathways to activate PMS. Similar to the Co-Fe/$SiO_2$ LC-PMS system, $Co^{3+}/Co^{2+}$ and $Fe^{3+}/Fe^{2+}$ took part in the $CoFe_2O_4$-SAC/PMS system to produce $SO_4\bullet^-$ and $\bullet OH$ as well.

$MnFe_2O_4$ as one of the spinel ferrite nanoparticles also has a superior catalytic activity, and simultaneously can further avoid the risk of Co leaching. Li et al. [56] synthesized sewage sludge-derived magnetic nanocomposites ($MnFe_2O_4$-SAC) to activate PDS for the degradation of Orange G. Likewise, $MnFe_2O_4$-SAC also displayed a synergistic function. More importantly, only moderate activation energy was required for Orange G degradation. In the meantime, both $CoFe_2O_4$-SAC and $MnFe_2O_4$-SAC showed excellent activation performance in neutral pH, which in turn embodied the potential for practical applications.

$$Co^{2+}/Fe^{2+} + HSO_5^- \rightarrow Co^{3+}/Fe^{3+} + SO_4\bullet^- + OH^- \tag{14}$$

$$Co^{2+}/Fe^{2+} + HSO_5^- \rightarrow \bullet OH + SO_4^{2-} + Co^{3+}/Fe^{3+} \tag{15}$$

$$Fe^{2+} + Co^{3+} \rightarrow Fe^{3+} + Co^{2+} \tag{16}$$

*4.2. Application in Fenton-like AOPs*

4.2.1. Fenton Oxidation Process

The heterogeneous Fenton process has been paid increasing attention since it degrades organic contaminants in a wider pH range and generates less iron sludge or less metal leaching than the homogeneous Fenton process [38]. For heterogeneous Fenton processes utilizing metal-containing catalysts, the reaction lies on the redox of the metal ions [78]. Recently research revealed that metal-free carbon catalysts, such as graphene oxide nanosheets [78] and multi-walled carbon nanotubes [79], provided a novel strategy for the prospective development of efficient and green Fenton-like catalysts. Thus, sludge-derived carbon catalysts can activate hydrogen peroxide and mediate the generating of ROS via their polyaromatic moieties, functional groups, the iron or other transition-metal impurities [62]. Other components including $SiO_2$, $Al_2O_3$ and carbon can act as efficacious promoters in heterogeneous Fenton-like reactions as well [80]. Catalytic activity of different sludge biochar-based catalysts in the Fenton oxidation process is concluded in Table 3.

**Table 3.** Application of sludge-based catalysts in Fenton-like AOPs.

| Synthesis Process | Product | Solution pH | Removal Capacity (RC) | Reusability and Chemical Stability | Mechanism | References |
|---|---|---|---|---|---|---|
| Sodium lauryl sulfate + sludge biochar + Kaolin were sintered at 1100 °C for 30 min under $N_2$. | SBC | 4.0 | 0.2 g/L product; 10 mg/L ciprofloxacin; 60 mmol/L $H_2O_2$. RC(ciprofloxacin) > 80% | - | $\bullet OH$ | [81] |
| The alkaline activated sludge was treated by microwave digestion and pyrolyzed at 600 °C for 2 h in $N_2$. | PFC600 | 5.0 | 0.5 g/L product; 1.0 mmol/L 1,2,4-Acid; 15 mmol/L $H_2O_2$. RC(1,2,4-Acid) = 96.6% | RC(1,2,4-Acid) still reached 90.2% after three recycles. | $\bullet OH$ | [82] |

**Table 3.** *Cont.*

| Synthesis Process | Product | Solution pH | Removal Capacity (RC) | Reusability and Chemical Stability | Mechanism | References |
|---|---|---|---|---|---|---|
| Sludge was activated by $H_2O_2$ at acid pH and then mixed with $FeSO_4 \cdot 7H_2O$. The resulting sample was carbonized at 600 °C under $N_2$. | SC-F-0.2 | 3.0 | 1 g/L product; 1 mmol/L Black-T; 20 mmol/L/L $H_2O_2$. RC(TOC) = 71% | The catalyst presented 2.77% of the iron load loss. The dye removal reached 91% after three repeated reactions. | - | [83] |
| The dry sludge was carbonized at 600 °C for 4 h under $N_2$ and then treated with sulfuric acid. | SC-$H_2SO_4$ | 6 | 1 g/L product; 30 mg/L ofloxacin; 138 mg/L $H_2O_2$. RC(ofloxacin) = 91.5% | - | •OH | [57] |
| The iron sludge was calcined at 600 °C for 3 h. | Fe-600 | 5.44 | 1 g/L product; 10 mg/L RhB; 10 mmol/Lol/L $H_2O_2$. RC(RhB) = 99% | - | •OH | [84] |
| The ferric sludge + biosolids mixture was stirred and sealed at 200 °C for 5 h. | SBMC | 3 | 1 g/L product; 0.21 mmol/L aniline; 60 mmol/L $H_2O_2$. RC(aniline) = 77.9%; RC(TOC) = 50.2% | In the 5th run, the catalytic ability of SBMC began to decrease and the leached iron was <0.75 mg/L. | •OH, •$O_2^-$ | [44] |
| The sludge + starch mixture was heated at 600 °C for 3 h under $N_2$ (SC). Then some SC was soaked with $H_2SO_4$ and the green tea extract was added. | SC-based catalyst | 4.0 | 1 g/L product; 20 mg/L NOR; 10 mg/L $Cu^{2+}$; 1.5 mmol/L $H_2O_2$. RC(NOR) = 98.8%; RC($Cu^{2+}$) = 97.5% | After four runs, RC($Cu^{2+}$) decreased from 97.5 to 39.1%, RC(NOR) decreased from 98.8 to 76.4%. | •OH (NOR), •$O_2^-$ ($Cu^{2+}$) | [58] |
| The dry Fe-rich sludge was calcined at 800 °C for 2 h in $N_2$. | Fe-SC-800 | 8 | 2 g/L product; AOII; 17 mmol/L $H_2O_2$. RC(AOII) = 98% | Fe-SC-800 performed similar RC after being recycled for three times. | •OH | [53] |
| WAS adsorbing the heavy metals was anaerobic pyrolyzed at 600 °C for 1 h. | Cu(II)-SBC | - | 0.1 g/L product; 800 µg/L E2; 600 mg/L $H_2O_2$. RC(E2) = 100% | - | •OH, •$O_2^-$ | [85] |
| | Ni(II)-SBC | | 0.1 g/L product; 800 µg/L E2; 600 mg/L $H_2O_2$. RC(E2) = 79% | | •$O_2^-$ | |
| Obtained through a co-precipitation method followed by sintering at 800 °C | $NiFe_2O_4$ | 3.0 | 2.0 g/L product; 250 mg/L phenol; 120 mmol/Lol/L $H_2O_2$. RC(phenol) = 95% | The leached iron amounted to 6.3% ± 0.2% of total iron. | •OH | [86] |
| Steel sludge was acid washed with HCl. | SS_HCl | 4.4 | 1 g/L product; 200 mg/L 4-CP; 20.3 mmol/L $H_2O_2$. RC(TOC) = 64% | The catalyst can be reused without any regenerative treatment for up to 3 cycles. | •OH, •$O_2^-$ | [15] |
| The sludge + $(NH_4)_2Fe(SO_4)_2$ mixture was calcined at 350 °C for 3 h in air. | FAS | 4.0 | 0.3 g/L product; 55.5 mg/L RhB; 3% $H_2O_2$. RC(TOC) = 69% | No obvious deactivation was observed over six repetitive trials. | •OH | [87] |

### 4.2.2. Fe-Based Catalysts

A number of studies have shown that many refractory organic pollutants such as antibiotic ciprofloxacin [81] and 1,2,4-Acid [82] can be degraded by sewage sludge containing Fe catalysts with $H_2O_2$. On the other hand, some pollutants such as dye anthraquinone [88], MB [89], and even real coal gasification wastewater [90] can be effectively treated by Fe-loading sewage sludge catalysts with $H_2O_2$. Even so, whether sludge containing Fe or Fe-loading sludge catalysts, it is normally necessary to take some measures in the preparation process for better catalytic performance.

To enhance iron insertion rate and catalyst stability, Wen et al. [83] prepared sewage sludge-derived Fe/carbon catalyst (SC-F) via radical pretreatment on sludge precursors. In comparison to direct iron impregnation, SC-F featured higher and well-dispersed iron nanoparticles tightly encapsulated in a carbon matrix, which gave rise to its better catalytic activity and stability in the heterogeneous Fenton process. Additionally, acid modification for sludge biochar-based catalysts is a feasible solution to boost up the catalytic performance and applicability of wider solution pH range in heterogeneous Fenton systems. Yu et al. [57] prepared sludge-derived carbon (SC) modified by sulfuric acid catalyst (SC-$H_2SO_4$) for heterogeneous Fenton degradation of ofloxacin. They found that the lower pH on the surface of SC-$H_2SO_4$ than on the solution was formed due to sulfate group, so that SC-$H_2SO_4$ still possessed superior catalytic activity under the wide range of solution pH (3–6). In particular, graphene-modified iron sludge catalyst [91] significantly promoted the

heterogeneous Fenton activity in view of the mesoporous structure as well as the electrostatic attraction and Π-Π conjugation, and meanwhile, the splendid electron conductivity of graphene was also beneficial for catalytic activity. Beyond these solutions, Guo et al. [84] stated that the self-doped sulfur in iron sludge could facilitate the electron transfer between the peroxide species and iron ions. This could accelerate the heterogeneous Fenton process for the degradation of rhodamine B (RhB). More importantly, the catalytic activity of the as-prepared catalyst (Fe-600) remained steady in the pH range of 3.20–10.25.

As far as Fe species were concerned, previous reports indicated that $Fe_3O_4$ [92] and ZVI [93] can be used as outstanding heterogeneous catalysts in place of Fe(II) in Fenton oxidation. On one hand, reductants are generally added for the preparation of sludge supported $Fe_3O_4$ or ZVI catalysts. Zhang et al. [44] made sludge-based magnetite catalyst (SBMC) containing $Fe_3O_4$ via hydrothermal process, in which the ferric sludge and biosolids were used as the feedstock. $Fe^{3+}$ reduction and the $Fe_3O_4$ synthesis mechanism were explored by clarifying the contribution of two main compositions of biosolids (protein and carbohydrate). In the end, a part of carbohydrate and the produced Maillard reaction products (MRPs) derived from two model substrates via Maillard reaction (MR) domination. The MRPs of fraction with molecular weight 50–100 kDa presented the highest relative reducing and chelation activity for $Fe_3O_4$ formation. Therefore, using SBMC as a Fenton-like catalyst for aniline degradation could achieve 77.9% aniline removal and 50.2% TOC removal under optimal conditions. In addition to biosolids, Liu et al. [58] adopted the green tea extract as reductive agent, thereby zero-valent iron ($Fe^0$) and zero-valent aluminum ($Al^0$) particles were formed on the sewage sludge-derived char-based catalyst (SC-based catalyst) surface. In the Fenton oxidation process, SC-based catalysts exhibited excellent performance on the simultaneous removal of $Cu^{2+}$ and NOR. For NOR degradation, it was primarily attributed to generated •OH by $Fe^0$ and $Al^0$ (Equations (17)–(20)). On the other hand, as previously discussed, Fe species in sludge could be reduced to $Fe_3O_4$ and $Fe^0$ without extra reductant at higher carbonized temperature via a series of reduction reactions in the oxygen-free pyrolysis process, and the resultant sludge-derived catalysts could take high catalytic activity in wide pH operating range [53]. Actually, more complicated phase evolution will occur during pyrolysis and then induce the formation of multivalent iron compounds.

$$2Fe^0 + O_2 + 4H^+ \rightarrow 2Fe^{2+} + 2H_2O \tag{17}$$

$$Fe^{2+} + H_2O_2 \rightarrow Fe^{3+} + \bullet OH + OH^- \tag{18}$$

$$2Al^0 + 3O_2 + 6H^+ \rightarrow 2Al^{3+} + 3H_2O_2 \tag{19}$$

$$Al^0 + 3H_2O_2 \rightarrow Al^{3+} + 3\bullet OH + 3OH^- \tag{20}$$

### 4.2.3. Other Metals-Based Catalysts

Extracellular polymeric substances (EPS) in waste activated sludge (WAS) have strong binding capacity with heavy metals, hence, Ai et al. [85] used WAS as an adsorbent for Cu(II) and Ni(II) removal. When WAS containing Cu(II) or Ni(II) was fabricated to sludge-based carbons (SBC) through anaerobic pyrolysis and served as heterogeneous Fenton carbon-based catalysts, electron-spin resonance spectroscopy (ESR) analysis found that both •OH and $\bullet O_2{}^-$ were devoted to 17β-estrogen (E2) removal in the Cu(II)-SBC-$H_2O_2$ system, while $\bullet O_2{}^-$ contributed to E2 degradation in the Ni(II)-SBC-$H_2O_2$ system. Besides, Cu(I) and Cu(0) were formed in Cu(II)-SBC catalytic process, whereas only Ni(II) was detected in Ni(II)-SBC catalytic process in the existence of $H_2O_2$. All in all, Cu(II)-SBC performed better catalytic activity in the heterogeneous Fenton process.

The spinel ferrite nanoparticles can activate persulfate as well as $H_2O_2$. Unlike the above-mentioned preparation (loading spinel ferrite nanoparticles on SAC), $NiFe_2O_4$ was obtained using Fenton sludge and $Ni(NO_3)_2$ as feedstock through a co-precipitation method followed by sintering [86]. It is interesting that in the Fenton system catalyzed by spinel, rapid electron exchange between Ni(II) and Fe(III) ions in the spinel structure

would prompt Fe(II) production and trigger more hydroxyl radicals, and subsequently eliminate phenol (95%). Afterward, the group synthesized $Cu_2O\text{-}CuFe_2O_4$ microparticles from Fenton sludge to degrade the same concentration of phenol [94]. Under the same catalyst dosage and less $H_2O_2$ conditions, it ended up with higher removal efficiency (97.2%). Similar to other AOPs, a higher degradation efficiency resulting from abundant free radicals is universally assigned to multiple redox pairs in this circumstance.

### 4.2.4. Photo-Fenton Process

Based on the heterogeneous Fenton process, the heterogeneous photo-Fenton process can use the energy provided by ultraviolet or visible light to further accelerate the reduction of $Fe^{3+}$ to $Fe^{2+}$ and enhance the abatement of organic pollutants (Table 3) [95]. For instance, steel industry sludge was prepared using acid washing as a catalyst (SS_HCl) for the degradation of 4-chlorophenol (4-CP) [15]. A comparative experiment of Fenton oxidation without UV irradiation was conducted. Results showed that within 2 h, the TOC removal for 4-CP reached 30% at an initial reaction pH of 4.4 with stoichiometric $H_2O_2$ dose. Meanwhile, about 60% $H_2O_2$ was consumed in the SS_HCl/$H_2O_2$ system, which can be ascribed to homogeneous (Equations (21) and (22)) and heterogeneous Fenton reactions (Equations (23) and (24)). By contrast, the TOC removal and $H_2O_2$ consumption surged to 64% and 90%, respectively, once UV irradiation was added. It was generally believed that the generated $Fe^{3+}$ irons could also convert back into $Fe^{2+}$ due to UV radiation and stimulate the production of hydroxyl radicals (Equation (25)) in the photo-Fenton system. In spite of that, only 4–5% of solar energy pertains to UV light, and visible light accounts for about 45% of the energy [96]. Satisfactorily, previous studies verified that sewage sludge-based materials still can be used as effective heterogeneous photo-Fenton catalysts under low-cost visible light irradiation conditions [97,98].

$$Fe^{2+} + H_2O_2 \rightarrow Fe^{3+} + \bullet OH + OH^- \tag{21}$$

$$Fe^{3+} + H_2O_2 \rightarrow FeOOH^{2+} + H^+ \rightarrow Fe^{2+} + HOO\bullet + H^+ \tag{22}$$

$$Fe^{2+}_{surface} + H_2O_2 \rightarrow Fe^{3+}_{surface} + \bullet OH + OH^- \tag{23}$$

$$Fe_3O_4 + H_2O_2 \rightarrow Fe^{3+}_{surface} + \bullet OH + OH^- \tag{24}$$

$$Fe^{3+} + OH^- + h\nu \rightarrow Fe^{2+} + \bullet OH \tag{25}$$

Given the influence of various iron salt doping, Yuan et al. [87] fabricated different digested sludge-based catalysts using $(NH_4)_2Fe(SO_4)_2$, $FeSO_4$, $FeCl_3$ and $Fe(NO_3)_3$ for the photo-Fenton reaction. As result, $Fe^{2+}$-loaded catalysts as heterogeneous photo-Fenton promoters were superior to $Fe^{3+}$-loaded catalysts. This phenomenon is ubiquitous in Fenton-like processes, mainly owing to the much slower reduction of $Fe^{3+}$ to $Fe^{2+}$. Although some advances were made, more efforts were needed to prepare sludge-based catalysts to simultaneously respond to external illumination and $H_2O_2$ activation in the photo-Fenton system.

### 4.3. Application in Photocatalysis

The combination of sludge and photocatalysts will bring unexpected results for wastewater treatment. The applications of different sludge biochar-based catalysts in photocatalysis are presented in Table 4.

**Table 4.** Application of sludge-based catalysts in photocatalysis.

| Synthesis Process | Product | Solution pH | Removal Capacity (RC) | Reusability and Chemical Stability | Mechanism | References |
|---|---|---|---|---|---|---|
| The sludge + $TiO_2$ mixture was carbonized at 200 °C for 20 h. Finally, the dried catalyst was heated at 800 °C for 2 h under $N_2$. | $TiO_2$ nanorods | 7 | 0.4 g/L product; 10 mg/L pentachlorophenol; RC(pentachlorophenol) = 97% | - | The photogenerated electrons | [99] |
| The sludge + HCl + $TiOSO_4 \cdot 2H_2O$ mixture was heated at 150 °C for 12 h. Finally, the dried solid was calcined at 700 °C for 5 h. | SS-Ti-700 | - | 35 mg/L p-NP; RC(p-NP) = 92.87% | SS-Ti-700 maintained excellent photoactivity in the six repeated experiments. | The photogenerated electrons, superoxide radical, and •OH | [100] |
| The pH of the sludge was adjusted to 1 and the titanium isopropoxide + sludge mixture was kept in a thermostat water bath for 6 h. The resulting precipitations were calcined in air at 500 °C for 4 h. | WSCT powder | 5 | 1 g/L product; 10 mg/L RhB; RC(RhB) = 82.4% | WSCT still exhibited high photo-activity after four times. | - | [101] |
| Titanium(IV) butoxide + ethanol + $HNO_3$ + sludge were mixed. After gelation of the sol, the product was heated at 200 °C for 2 h. | $TiO_2$/ASS nanocomposite | 7 | 4 g/L product; 25 mg/L MO; 30 mg/L $Cd^{2+}$; RC(MO) = 94.28%. RC($Cd^{2+}$) > 90% | - | - | [102] |
| The sludge + $TiO_2$ + NaOH mixture was heated at 200 °C for 20 h. The dried samples were heated at 600 °C for about 0.5 h under $N_2$. | 0.01 g/L SS-$TiO_2$ (ST2) | - | 0.01 g/L product; 5 mg/L tetracycline; RC(tetracycline) = 76.3% | - | - | [23] |
| The zinc acetate + SBAC + NaOH mixture was maintained at 180 °C for 12 h. | ZnO-SBAC (ZC) | 2.35 | 1 g/L product; 10 mg/L Cr(VI); RC(Cr) = 93.61% | RC(Cr) was similar after 3 cycles. | - | [103] |
| The RSDBC + zinc nitrate mixture was treated at 450 °C for 3 h under Ar. | ZnO@RSDBC | 7 | 0.9 g/L product; 20 mg/L AO7; RC(AO7)~95% | RC(AO7) remained at 78.6% after three cycling runs. | $h^+$, $O_2 \bullet^-$, $SO_4 \bullet^-$ and HO• | [10] |

### 4.3.1. $TiO_2$-Based Catalysts

Due to low cost, non-toxicity, and photochemical stability, $TiO_2$ is one of the most promising photocatalysts and has received widespread attention in recent years [104,105]. However, the photocatalytic activity of $TiO_2$ is limited by two main drawbacks, including its wide band gap of about 3.0–3.2 eV that causes only UV light energy to be used, and its low adsorption of contaminants in solution [99].

Considering organic and inorganic ingredients, such as silica and transition metals in sewage sludge, the combination of $TiO_2$ and sewage sludge may solve these problems. For example, Matos et al. [106] reported that the presence of iron phases, mainly iron oxides and carbides obtained from municipal sewage sludge photos, assisted the $TiO_2$ in the photodegradation of MB, in agreement with results from Yuan et al. [100]. Yuan et al. [100] found that *p*-nitrophenol (*p*-NP) can be effectively degraded (92.87%) and mineralized (47.28%) over sludge-supported $TiO_2$ (SS-Ti-700) catalyst under the visible light irradiation, while the *p*-NP removal efficiency was 16.24% for the pure $TiO_2$. For the SS-Ti-700 under illumination, the 3d electrons of the in situ metal-ion doping can be excited and injected from the excited impurity level of these doped metal ions, mainly $Fe^{3+}$, to the conduction band of the $TiO_2$. Then $O_2$ can react with the injected photogenerated electrons and generate •OH (Equation (26)). In the presence of Cr(VI) as the electron scavenger, the generation process of •OH were given as Equations (27) and (28). Finally, the photocatalytic reaction mechanism of $TiO_2$-based photocatalysts using sludge as both the scaffold template and dopant was proposed (Figure 4). Likewise, sludge templated $TiO_2$ photocatalyst (WSCT) displayed a synergistic effect between sludge and $TiO_2$ by the photodegradation of RhB and methyl orange (MO) [101], respectively. Ultraviolet-visible-near infrared diffuse reflectance (UV-vis-NIR) spectra, ESR spectra and photoluminescence (PL) emission spectra showed that WSCT possessed higher light absorption capacity and electron-hole pair separation efficiency, showing better photocatalytic activity compared with naked $TiO_2$.

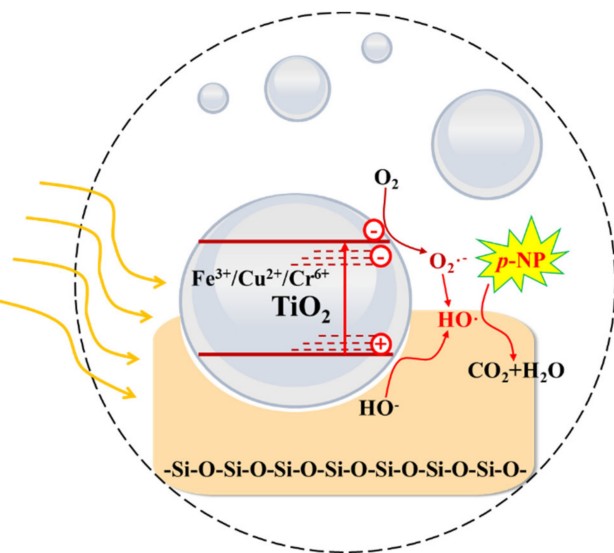

**Figure 4.** Mechanism of photocatalysis under visible light irradiation [100].

Even bi-pollutant solutions such as MO and $Cd^{2+}$ can be simultaneously removed by adsorption and photocatalysis of the sludge-supported $TiO_2$ catalyst [102]. Moreover, in the $TiO_2$/sludge-based catalysts synthesis process, the calcination gasses, such as $N_2$ and air have an evident influence on properties of the photocatalysts. Zhu et al. [23] successfully synthesized municipal sewage sludge and combined $TiO_2$ photocatalysts under air ($ST_1$) and $N_2$ ($ST_2$) atmospheres for photocatalytic degradation of tetracycline. According to characterization results, more non-metal elements were found in the $ST_2$, and it had higher BET surface area (92.97 $m^2$/g), due to its carbonaceous graphene-like structure, when compared to $ST_1$ (73.78 $m^2$/g) and $TiO_2$ (42.45 $m^2$/g). Consequently, $ST_2$ (76.3%) showed a higher removal efficiency of tetracycline than $ST_1$ (56.8%) under visible light irradiation. The enhancement was formed together by a large organic surface structure, the transition metals and carbon dopants of $ST_2$, hence the adsorption of tetracycline and visible light was increased and electron-hole pair separation was accelerated.

$$O_2 + e^- \rightarrow O_2 \bullet^- \qquad (26)$$

$$h^+ + OH^- \rightarrow \bullet OH \qquad (27)$$

$$h^+ + H_2O \rightarrow \bullet OH + H^+ \qquad (28)$$

Notably, the additional doping of N in the biochar-based $TiO_2$ can significantly reduce band gap energy of composite through the formation of a mid-gap energy state or local trap state between $TiO_2$ bands and hence promote the photodegradation of pollutants [107]. Thus, the N-doped sludge-based $TiO_2$ catalysts need to be further explored.

### 4.3.2. ZnO-Based Catalysts

In addition to $TiO_2$, ZnO is also widely used in photocatalysis [105]. Ramya et al. [103] employed tannery sludge activated carbon (SBAC) supported ZnO nanocomposites (ZnO-SBAC) for the photoreduction of Cr(VI). Similar to other biomass carbon-supported ZnO catalysts, e.g., the jute fiber carbon-based ZnO, its photocatalytic property depends on the ratios of raw materials. Similarly, it was also vital to ZnO-SBAC. The composite at weight ratio of 2:3 among the all prepared weight ratios of ZnO to SBAC was found to have the lowest band gap energy of 2.96 eV and highest photocatalytic activity, wherein the introduction of sludge activated carbon prevented the $e^-$/$h^+$ recombination process by trapping the photogenerated electrons (Figure 5). In addition, a novel core-shell structure photocatalyst was obtained by loading ZnO on rectorite/sludge-derived biochar (ZnO@RSDBC) [10].

Compared with ZnO, the photocatalytic performance of ZnO@RSDBC was enhanced due to the formation of a heterogeneous interface.

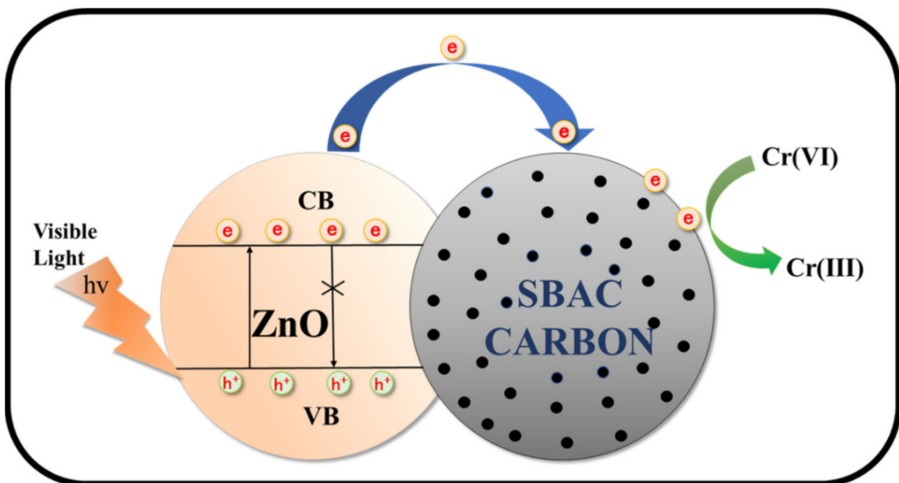

**Figure 5.** Mechanism for the photocatalytic reduction of Cr(VI) on the surface of ZnO-SBAC under visible light irradiation [103].

### 4.3.3. Graphitic Carbon Nitride-Based Catalysts

The g-$C_3N_4$ is an emerging material for the degradation of pollutants owing to its high thermal stability (600 °C in air), tunable electronic band gap (~2.7 eV), metal-free composition, and nontoxicity [108]. Sludge as a supporting matrix of g-$C_3N_4$ can improve its photocatalysis and adsorption capacity, like a composite material of g-$C_3N_4$ from melamine and alum sludge obtained by Kim et al. [109]. During the synthesis, alum sludge and melamine mixed in different mass ratios were placed in a muffle furnace under a nitrogen environment, and the reactor was heated to 550 °C and maintained for 4 h. It should be noted that g-$C_3N_4$ photocatalysts cannot adsorb any species of arsenic; by contrast, the resulting composite can photocatalytic oxidize As(III) to As(V) and adsorb As simultaneously under the light.

### 4.4. Application in Ozonation

Heterogeneous catalytic ozonation using sludge biochar-based catalysts can improve the efficiency of the ozonation process and provide more effective mineralization of many organic pollutants (Table 5).

**Table 5.** Application of sludge-based catalysts in ozonation.

| Synthesis Process | Product | Solution pH | Removal Capacity (RC) | Reusability and Chemical Stability | Mechanism | References |
|---|---|---|---|---|---|---|
| Pyrolyzed at 700 °C for 2 h under $N_2$. | Biochar | - | 1.0 g/L product; 0.2 g/L phenol; 14 ± 1 mg/L and 1.0 L/min $O_3$. RC(phenol) = 95.4% | RC(phenol) dropped to 59.3% (fourth trial). | $\bullet O_2^-$ | [21] |
| Activated by $ZnCl_2$/KOH/$H_2SO_4$ and then pyrolyzed at 700 °C for 1 h under $N_2$. | SBC | 4.0 | 0.2 g/L product; 0.1 mmol/L oxalic acid; 0.7 mg/min $O_3$. RC(oxalic acid) = 81.2% | - | Surface reaction | [33] |
| The SAC + $FeCl_3 \cdot 6H_2O$ + $FeSO_4 \cdot 7H_2O$ mixture was put into the thermostat water bath at 93 °C for 3 h. | FMSAC | 6.0 | 0.04 g/L product; 20 mg/L p-CBA; 1 mg/L $O_3$. RC(p-CBA) = 80% | RC(p-CBA) only reduced by 13.2% after six repetitive runs. | $\bullet OH$ | [110] |
| SAC was obtained by $ZnCl_2$+$H_2SO_4$ activation and then pyrolyzed at 550 °C for 1 h under $N_2$. The SAC + manganese mixture was calcinated at 550 °C for 1 h under $N_2$. | $MnO_x$/SAC | 3.5 | 0.1 g/L product; 80 mg/L oxalic acid; 5.0 mg/L $O_3$. RC(oxalic acid) = 72.1%; RC(TOC) = 92.2% | Manganese leaching was approximately 3.9% in 60 min. | Surface reaction was dominant | [34] |

**Table 5.** *Cont*.

| Synthesis Process | Product | Solution pH | Removal Capacity (RC) | Reusability and Chemical Stability | Mechanism | References |
|---|---|---|---|---|---|---|
| SCCA-Zn was obtained by ZnCl$_2$ activation and then pyrolyzed under N$_2$. SCCA-Zn + MgSO$_4$ + NaOH were mixed. Then precipitation was squeezed into pellets and finally thermal treated at 500 °C for 2 h under N$_2$. | Granular MgO-SCCA-Zn | 9.0 | 10 g/L product; 500 mg/L MB; 5 mg/L O$_3$. RC(MB) = 98%; RC(COD) = 51.12% | Mg$^{2+}$ was not detected in the aqueous solution. After the 3th reuse, RC(COD) was 49.68%. | - | [111] |
| SBC$_0$: sludge was pyrolyzed at 850 °C for 1 h under N$_2$. SBC$_a$: SBC$_0$ was treated with HCl solution. SBC$_b$: SBC$_a$ was treated with NaOH solution. | Activated petroleum waste sludge biochar | 7.7 | 1.0 g/L product; petroleum refinery wastewater; 20 mg/min O$_3$. The SBC$_0$-COP (53.5%), SBC$_a$-COP (49.3%) and SBC$_b$-COP (51.8%) all exhibited higher TOC removal | After the 5th reuse, RC(TOC) was reduced to 48.5% (SBC$_0$), 37.9% (SBC$_a$) and 37.3% (SBC$_b$). | •OH | [22] |
| SBAC: activated by ZnCl$_2$ and then pyrolyzed at 700 °C for 1 h under N$_2$. The SBAC + Mn nitrate mixture was calcinated at 600 °C for 3 h under N$_2$. | MnO$_x$/SBAC | 6.5–7.5 | 1 g/L product; coal gasification wastewater; 500 ml/min and 15 mg/L O$_3$. RC(COD) = 78.1% | The maximum leaching of Mn was 0.41 mg/L. RC(COD) kept higher than 63.2% throughout ten successive runs. | •OH | [59] |
| SBAC: activated by ZnCl$_2$ and then pyrolyzed at 700 °C for 1 h under N$_2$. The SBAC + Fe nitrate mixture was calcinated at 600 °C for 3 h under N$_2$. | FeO$_x$/SBAC | 6.5-7.5 | 1 g/L product; coal gasification wastewater; 500 ml/min and 15 mg/L O$_3$. RC(COD) = 73.7% | The maximum leaching of Fe was 1.45 mg/L. RC(COD) kept higher than 63.2% throughout ten successive runs. | •OH | [59] |
| Pyrolyzed at 700 °C for 4 h under N$_2$. | SC-700 | 6 | 0.5 g/L product; 200 mg/L HQ; 50 ml/min and 17 mg/L O$_3$. RC(HQ) = 97.86% | - | - | [112] |
| The sewage sludge + corncob + ZnCl$_2$ mixture was pyrolyzed at 600°C for 1h. | SCAC | 6.0 | 0.025 g/L product; 0.5 mg/L Ibuprofen (IBP); 3.0 mg/L O$_3$. RC(IBP) = 100% | - | •OH | [113] |
| MSAC: sludge was activated by ZnCl$_2$ and then pyrolyzed under N$_2$. The SAC + Fe(NO$_3$)$_3$·9H$_2$O + HNO$_3$ mixture was pyrolyzed at 600 °C for 1 h under N$_2$. For the Mn loaded MSAC, the molar ratio of Fe to Mn was adjusted to 2. | MSAC-Mn | 7.0 (phosphate buffer) | 0.05 g/L product; 0.6 mg/L IBP; 1.0 mg/L O$_3$. RC(IBP) = 86.2% | - | •OH | [114] |

On one hand, pure sludge-based biochar can catalyze O$_3$ to form ROS and then degrade pollutants. Zhang et al. [21] prepared biochars at 300, 500, 700, and 900 °C pyrolysis temperatures, which were denoted as BC300, BC500, BC700, and BC900. The catalytic activity of biochars towards ozonation of phenol was evaluated, and the results showed that 95.4% of phenol removal could be reached by BC700 or BC900, however, BC500 revealed lower phenol removal (89.3%). The superior catalytic activity of BC900 and BC700 contributed to a higher percentage of carbonyl groups that can transfer electrons to dissolved ozone to form superoxide radicals (Figure 6). Additionally, changing initial pH had little influence on phenol removal. Wen et al. [33] found that oxalic acid removal with sludge-based carbon was 81.2% after 40 min under initial pH 4 and the removal efficiency of oxalic acid decreased as initial pH increased. This may be related to the pHpzc of the catalyst [115].

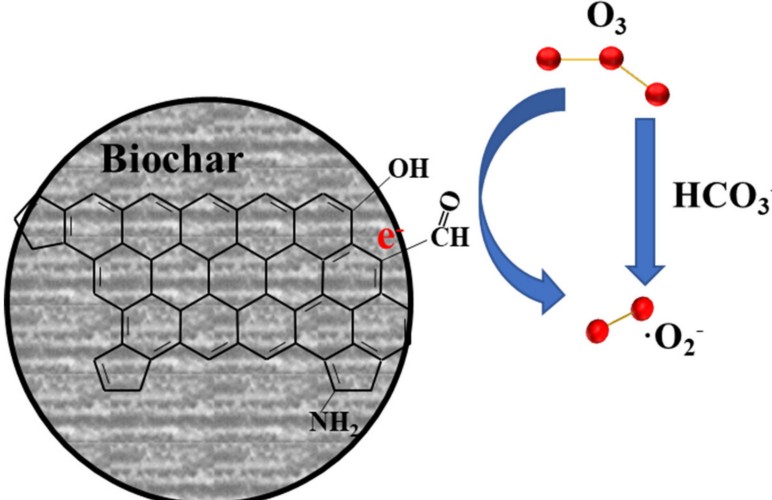

**Figure 6.** Mechanism of ozonation catalyzed by biochar [21].

On the other hand, a combination of sludge with metal oxides can further improve catalytic ozonation capacity. Lu et al. [110] synthesized ferromagnetic sludge-based activated carbon (FMSAC) and analysis indicated that magnetite and maghemite were the main magnetic phases, so that the prepared FMSACs could be easily separated by magnetic fields. In the initial period of catalytic ozonation, the presence of FMSACs can obviously enhance p-chlorobenzoic acid (p-CBA) removal, and the quenching experiments confirmed indirectly the transformation of ozone into •OH. As for transition metal manganese, Dhandapani and Oyama [116] had already proposed that MnOx were the most active catalyst for gas phase ozone decomposition oxide due to the generation of peroxide species among some transition metals and noble metals. In aqueous phase, Mn and its multivalence can lead to electron transfer between metal oxides and $O_3$ and promote the generation of •OH [34], which was described in Figure 7.

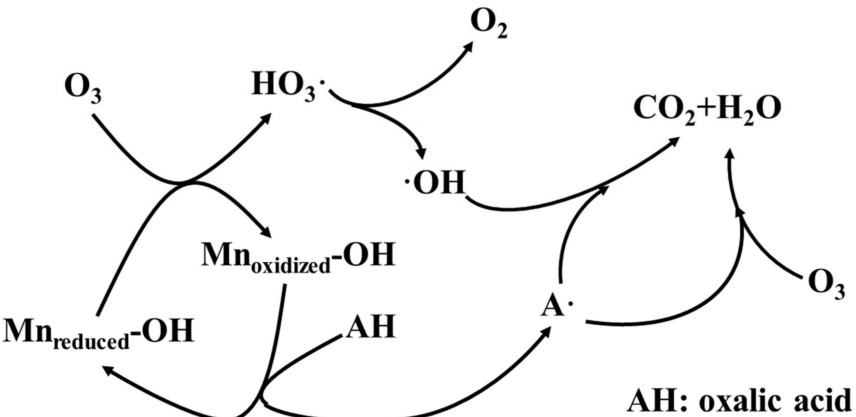

**Figure 7.** Mechanism of electron transfer between Mn and $O_3$ [21].

Except for transition metal oxides, MgO as an efficient basic ozonation catalyst similarly takes preferable catalytic activity. Kong et al. [111] pelletized the synthesized porous sludge-derived char (SCCA-Zn) and $Mg(OH)_2$ mixtures to prepare granular MgO-SCCA-Zn (G-MgO-SCCA-Zn) hybrid ozonation catalyst. G-MgO-SCCA-Zn performed not only well in water resistance but in catalytic potential degradation of methylene blue when being used in the catalytic ozonation process (COP). Compared to commercial granular AC (G-AC) and granular MgO (G-MgO), G-MgO-SCCA-Zn had a higher surface area and pore volume. More importantly, due to the nano-particle size of MgO (50 nm) on the surface of G-MgO-SCCA-Zn, the reaction rate in the existence of G-MgO-SCCA-Zn was higher than G-AC and G-MgO.

Use of waste sludge-derived catalysts in COP of real wastewater not only significantly enhances the performance of toxic pollutant removal but also improves profitability. Chen et al. [22] employed catalytic ozonation involving petroleum waste sludge biochar catalyst (SBC) for the treatment of refinery wastewater and found that the total number of polar contaminants was reduced by 39.6% (1379 vs. 2285) through ●OH-mediated oxidation, resulting in enhancing TOC removal efficiency compared to ozonation. An especially large portion of oxygen-containing and sulfur-containing compounds were removed by $SBC_0$-COP. Sludge could be also converted to sludge-based activated carbon (SBAC), which supported manganese ($MnO_x$/SBAC) and ferric oxides ($FeO_x$/SBAC) as catalysts to enhance catalytic activity in the ozonation of biologically pretreated coal gasification wastewater that contained considerable amounts of toxic and refractory compounds [59]. In this system, the average effluent concentrations of COD, TP and TOC in ozonation with $MnO_x$/SBAC and $FeO_x$/SBAC were 33–39, 0.6–1.5 and 17–18 mg/L, respectively. Meanwhile, the $BOD_5$/COD value increased gradually from 0.06 to 0.55 and the toxicity inhibition decreased by 4.0–5.0 times, which was beneficial to the subsequent treatment, such as the anoxic moving bed biofilm reactor (ANMBBR) and the biological aerated filter (BAF) process [54]. We can also learn from Table 5 that sludge-based catalysts in catalytic ozonation of real wastewater revealed superior stability.

## 5. Costs and Viability

Due to the efficient and economic characters, sludge biochar-based catalysts proved to be promising heterogeneous advanced oxidation for contaminants. In recent years, they play a key role in real wastewater treatment including real biologically pretreated coal gasification wastewater and refinery wastewater. The sludge biochar-based catalysts not only significantly enhanced the performance of toxic pollutants removal but also improved profitability. In general, the net margin of sludge biochar-based catalysts application could be increased by choosing cheaper dopants such as transitional metals (Fe, Mn), and by applying a promising processing technology such as slow pyrolysis. However, they are still confronted with the decrease of active sites such as the metal leaching. This propensity can be significantly reduced by enhancing solution pH. A significant number of sludge biochar-based catalysts have been prepared and show satisfactory performance in a wide range of pH. More importantly, the dilemma of being oxidized urgently needs to be solved. Thus, more efforts should be made in future work to improve the antioxidation of the sludge biochar-based catalysts in the heterogeneous advanced oxidation of the wastewater.

## 6. Conclusions and Prospects

### 6.1. Conclusions

Currently, the amount of sludge expands sharply, and sludge treatment has turned out to be an overwhelming issue around the world, so alternative uses for sludge are essential to develop. This review shows that the conversion of sludge into sludge biochar-based catalysts is a promising strategy that merges the merits of waste reutilization and environmental cleanup. On one hand, using some preparation methods such as pyrolysis and hydrothermal carbonization we are able to reduce raw sludge toxicity. On the other hand, considering components of the sludge itself, such as organic compounds and metals, the formation of OFGs and metal phases in preparation processes can act as active sites for pollution degradation.

The doping of metals and heteroatoms in most cases contributes to improving the catalytic performance of sludge biochar-based catalysts, and may alter oxidation pathways. Meanwhile, some modification methods, such as acid and graphene modification in preparation processes, are also favorable. Moreover, sludge as a supporting matrix is beneficial to enhancing catalyst property, reusability and stability. On this basis, pure sludge-based catalysts, metal-doped composites, heteroatom-doped composites and other types of catalysts are successfully fabricated and applied in different systems, including sulfate-based AOPs, Fenton-like AOPs, photocatalysis and ozonation. As a result, these

catalysts exhibit splendid catalytic performance for the abatement of contaminants. In spite of that, these systems are constrained by solution pH, catalyst dosage, reaction temperature and coexisting anions in general. The catalytic mechanisms of sludge biochar-based catalysts are also complex. Overall, radical pathways and nonradical pathways are devoted to catalytic processes.

*6.2. Prospects*

A great breakthrough in this field has been gained. Nevertheless, many improvements remain to be done in the future study of the large-scale practical application of wastewater treatment:

1. When sludge biochar-based catalysts are repeated several times, catalytic activity may diminish, which is ascribed to the decrease of active sites and defect intensity as catalysts surface encounters the dilemma of being oxidized. Therefore, future sludge-based catalysts synthesis should focus on effectual modification methods and feasible dopants in order to strengthen the reusability and stability of sludge biochar-based catalysts;
2. Few studies have concentrated on addressing the real wastewater treatment with sludge biochar-based catalysts in AOPs. Future investigations on catalytic efficiency in actual effluent are required;
3. The mechanism involved is quite complicated. Understanding the correlation between sludge biochar-based catalysts, different parts and real wastewater components such as hydrophilic and hydrophobic organic matter will give direction for the synthesis or modification of sludge biochar-based catalysts.

**Author Contributions:** Writing—original draft preparation, X.C. and L.F.; writing—review, C.W. and Y.C.; searching—literature, Y.Y., M.L., X.J., J.Y., and P.W. All authors have read and agreed to the published version of the manuscript.

**Funding:** The work was founded by the basic research fund of the central public-interest scientific institution (grant number: 2020YSKY-005) and China special S&T project on treatment and control of water pollution (grant number: 2017ZX07402002).

**Institutional Review Board Statement:** Not applicable.

**Informed Consent Statement:** Not applicable.

**Data Availability Statement:** In this paper, the data from Google Scholar, Web of Science, China National Knowledge Infrastructure, China Wan Fang Literature Database, and China WeiPu Literature Database.

**Conflicts of Interest:** The authors declare no conflict of interest.

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
