# Peer review of "Recent Development in Sludge Biochar-Based Catalysts for Advanced Oxidation Processes of Wastewater"

_catalysts, doi:10.3390/catal11111275_

Round 1
Reviewer 1 Report
This paper reviews an adequate number of papers dealing with the production and application of catalysts from sewage sludge in advanced oxidation processes.
This topic is of interest to the readers of this journal. The manuscript is well arranged and presents some recent findings. I recommend publication after minor revisions,
Please, consider the following comments:
- The manuscript requires a revision due to the presence of several typos and misuse of articles
- The text is most effective where data are arranged in tables. Please, consider introducing a table also for summarizing the data proposed in the paragraph 4
- The paragraph called methods (where the research principle is reported ) seems to me superfluous and should be removed.
- Please, avoid multiple citations (e. Zhang et al., 2019; Kwon et al., 2012; Zaker et al., 2019; Swann et al., 2017) by specifying why each paper was mentioned.
- Are the figures in the text original or are they extracted from other papers? In this case their reference should be included in the caption.
- A paragraph on costs and viability is missing
- Another missing aspect, which is of great interest and therefore worth investigating, is the possible formation or release of potentially dangerous compounds
Reviewer 2 Report
Dear Editor,
Regarding this paper, I have minor observations, in order to improve the overall quality of this good piece of work.
After that, I do recommend the publication of the manuscript.
Sincerely yours.
Numbering lines would facilitate the reading.
It is not clear the differences between “correspondence” (corresponding?) authors number 1, 2, and 3.
Abstract: replace “metals/nonmetals” to pollutants dopants. (there are metalloids as well).
Introduction:
“Since sludge contains many toxic substances”… many toxic is quite vague. Which ones?
Methods:
“Among all the cited papers, the earliest publication year of articles on the sludge-biochar based catalyst was 2011.” This is a result, not a method.
3.3 Dopants
“For sulfate-based AOPs, the doping of nitrogen can substantially alter catalytic oxidation pathway from a radical process to a nonradical process”. Why this sentence has been in underline form?
4.4. and in whole text:
Sometimes authors cite dosages in mg/L and in other places in mmol/L. It would be usefull if the paper standardize these units.
Table 1, 2, 3, and 4: The removal capacities should be reported in the same basis.
- Concluding remarks would be more clear as a heading of this session.
Congratulations.
